# Clinical Course, Genetic, and Immunohistochemical Characterization of Solid Pseudopapillary Tumor of the Pancreas (Frantz Tumors) in a Brazilian Cohort

**DOI:** 10.3390/genes13101809

**Published:** 2022-10-06

**Authors:** Francinne T. Tostes, Parisina Fraga Dutra Cabral de Carvalho, Raphael L. C. Araújo, Rodrigo Chaves Ribeiro, Franz Robert Apodaca-Torrez, Edson José Lobo, Diogo Bugano Diniz Gomes, Donato Callegaro-Filho, Gustavo Schvartsman, Fernando Moura, Vladimir Schraibman, Alberto Goldenberg, Fernanda Teresa de Lima, Vanderlei Segatelli, Pedro Luiz Serrano Uson Junior

**Affiliations:** 1Department of Oncology, Hospital Israelita Albert Einstein, São Paulo 05652-900, Brazil; 2Center for Personalized Medicine, Hospital Israelita Albert Einstein, São Paulo 05652-900, Brazil; 3Department of Surgery, Universidade Federal de Medicina do Estado de São Paulo, São Paulo 04023-062, Brazil; 4Department of Pediatric Surgery, Hospital de Cancer de Barretos Fundação PIO XII, Barretos 14784-370, Brazil; 5Barretos School of Health Sciences Dr. Paulo Prata—FACISB, Barretos 14784-370, Brazil; 6Department of Pathology, Hospital Israelita Albert Einstein, São Paulo 05652-900, Brazil

**Keywords:** Frantz tumor, solid pseudopapillary pancreatic neoplasm, pancreatic tumor, Li-Fraumeni syndrome

## Abstract

Frantz tumors or solid pseudopapillary pancreatic neoplasm (SPN) are rare exocrine neoplasms that carry a favorable prognosis; they represent up to 3% of all tumors located in the region of the pancreas and have specific age and gender predispositions. In recent years, the rising curve of diagnosis is entitled to the evolution and access of diagnostic imaging. In this paper, we have retrospectively reviewed and described the clinical course of 40 patients with SPN from three institutions in Brazil, who had their diagnosis between 2005 and 2020, and analyzed the clinicopathological, genetic, and surgical aspects of these individuals. In accordance with the literature, most patients were women, 60% with unspecified symptoms at diagnosis, with tumors mainly located in the body and tail of the pancreas, of whom 70% underwent a distal pancreatectomy with sparing splenectomy as a standard procedure, and none of the cases have experienced recurrence to date. Surgery still remains the mainstay of treatment given the low metastatic potential, but more conservative approaches as observed in this cohort are evolving to become the standard of care. Herein, we present an in-depth analysis of cases focusing on the latest literature and report some of the smallest tumor cases in the literature. To our knowledge, this is the first report evaluating germline genetic testing and presenting a case of detected Li-Fraumeni syndrome.

## 1. Introduction

Solid pseudopapillary pancreatic neoplasm (SPN), or Frantz tumor, is an uncommon neoplasm distinct from other neoplasms that originated in the pancreas. SPN typically has low metastatic potential and a very distinct behavior when compared to pancreatic adenocarcinomas [1,2,3]. It was initially described by Frantz in 1959, and since then cases have been described in the literature. It accounts for 1–3% of all pancreatic malignancies, and the overall tumor mortality rate is estimated to be around 2%. Although rare, a 7-fold increase in incidence since 2000 has been reported in English data, attributable probably to better awareness and access to imaging diagnosis. Generally, 90% of patients are female and 85% are under 30 years of age at diagnosis, with a median age of diagnosis of 26 years [2,3,4]. 

The tumor can occur anywhere in the pancreas and has a macroscopic presentation as a round, well-demarcated lesion, averaging 7–8 cm n length [2]. Surgery is the definitive treatment for almost all cases, with multiple different techniques with the objective of complete resection [5]. A range of different sizes is reported, from 2 cm to 34 cm; often, patients presented indirectly correlated symptoms [6].

Cases of metastatic disease are rare. Reports of metastatic dissemination include cases with local pancreatic recurrence, lymph nodes, and liver recurrences [7,8]. Most cases are treated with further resections with an excellent control rate and a high possibly of cure [7,8]. Systemic treatments, including chemotherapy, do not seem to have tumor effects [8,9]. Some reports in younger patients presented liver transplantation as an effective strategy for patients with synchronic multiple liver metastases [8,10,11].

In this manuscript, we report cases treated in our institutions. We address clinical, pathological, and genomics aspects of patients with SPN who underwent curative-intent resection. We also correlate our findings with the current mainstream management of the diagnosis and treatment of SPN.

## 2. Materials and Methods

We performed a retrospective review of 40 confirmed cases of solid pseudopapillary tumors. The data were collected from three tertiary hospitals in Brazil where the cases were conducted by the authors. The collection of data was carried out between March 2005 and December 2020. The group of eligible patients was described using standardized and prospectively databases from the departments of radiology, pathology, oncology, genetics, and epidemiology. All of the information was self-collected and conducted by the authors. Patients clinical characteristics, symptoms, follow up, surgery, pathology, and other relevant data were extracted from the medical records. Eligible patients were 10 years of age or older and had a confirmed histological diagnosis of solid pseudopapillary tumor of the pancreas. All of them underwent surgical resections.

Germline sequencing using a greater than 80-gene next-generation sequencing platform analysis was performed in 5 of these patients.

The discussion was based on a review of literature through a database search in Pubmed till December of 2020 including published articles about the subject. This case series was approved by national standards by the institutional review boards and ethics committees of the respective hospitals CAAE: 81744017.6.0000.0071, CAAE: 95748818.1.1001.5437, and CAAE: 31976520.5.0000.5505. 

### Patients

A total of 40 cases were included in this case series. Most patients were female (95%), and the median age was 17 years old (10–49). Forty percent of the patients presented with no symptoms, and those who had any presented with unspecified abdominal pain (57%). The most common surgery performed was distal pancreatectomy, realized in 23 (57%) patients. The size of tumors ranged between 0.9 cm and 15 cm (Table 1). In the cases in whom immunohistochemistry was available, all of them tested positive for CD99, a transmembrane protein, and β-catenin, a multifunctional protein. Both represent immunomarkers that, in combination, are a useful method in the diagnosis of SPN. Additionally, five patients, four females and one male, were genetically evaluated through the Cancer Genetics Clinics of one of the collecting database hospitals.

## 3. Results

### 3.1. Clinical Characteristics

In our data, we had six patients diagnosed and treated with more than 40 years old (Table 1). Three of them had abdominal pain, and the other three were asymptomatic. All of them were female. No clinical distinct characteristics were identified in this group of patients compared to most younger patients in this cohort. Although some of our colleagues relate age to patterns of failure and poor prognosis, none of them had recurrence or complications till the report of this series.

Regarding location, any site of the pancreas can be affected, with the majority of the tumors occurring at the pancreatic body and tail. One of our cases was a female patient, 41 years old and asymptomatic, who, to our knowledge, had the smallest tumor reported in the literature, measuring 0.9 cm. The tumor was identified in a routine abdominal ultrasound (Figure 1).

The two male patients from the cohort were 29 and 33 years old, respectively, at diagnosis. Both were asymptomatic at diagnosis, had higher ages then the average of the overall cohort (17 years old), and both were treated with distal pancreatectomy. Furthermore, no clinically relevant differences were noted in these patients. It is important to note that this observation agrees with the literature, where male patients are diagnosed with SPN at older ages than female patients; however, the prognosis remains the same.

### 3.2. Radiological Features

SPN can be diagnosed through a variety of methods, including ultrasonography, magnetic resonance imaging (MRI), computed tomography (CT), and positron emission tomography. Although rare, radiography can show possible calcifications associated with the neoplasm but with low potential for definitive diagnosis. Some cases as an example are identified with different imaging exams.

Case 1: A 41-year-old asymptomatic female patient underwent routine abdominal ultrasound where a solid, well-circumscribed 0.9 cm nodule was observed in the cephalic portion of the pancreas. Given the size of the tumor, no proper etiology could be defined prior to surgery (Figure 1). Magnetic nuclear resonance was performed, which confirmed the lesion. The patient was then treated with a pancreaticoduodenectomy.

Case 2: A 46-year-old patient presented with insidious abdominal pain. Abdominal ultrasound and tomography showed a solid expansive cystic lesion in the pancreas that measured approximately 7 cm, with no invasion of adjacent structures (Figure 2). Differential diagnosis could include pseudocysts, serous tumors, and mucinous tumors. The patient was treated with a gastroduodenopancreatectomy.

### 3.3. Surgical Aspects

The most common surgical procedure performed in this case series was distal pancreatectomy (with or without splenectomy) given the location of the majority of tumors occurring in the body and tail of the pancreas. The complete surgical excision provides curative-intent treatment for more than 95% of patients, even for locally advanced lesions. Although local invasions, recurrences, or limited metastases are not strict contraindications for resection and disease-free interval, vessel encasements and sites of metastasis play a decisive role in the decision-making process and the timing of surgery.

Although there is no consensus about the extension of the pancreatectomy for SPS, considering the low potential risk for metastatic spread perhaps justifies a more conservative approach with no radical lymph node resection as a standard treatment.

### 3.4. Anatomopathological Aspects

Most in this case series were solitary tumors in the pancreas, well delimited, with great variation in their size. They presented a heterogeneous surface, with solid areas, necrosis, and hemorrhagic cavities (Figure 3). Nowadays, with the diagnostic evolution of imaging tests and preventive medicine, these tumors are increasingly being diagnosed early. 

The SPN has distinct microscopic characteristics. Neoplastic cells have large round or oval nuclei, with ample eosinophilic or clear cytoplasm, often vacuolated. These tumors often have areas containing PAS-positive intracytoplasmic eosinophilic hyaline blood cells. The stroma has delicate vessels, with pseudopapillary areas (Figure 4A).

The immunohistochemical study is essential in some cases to rule out differential diagnoses, such as acinar cell carcinoma and pancreatoblastoma, but especially well-differentiated neuroendocrine tumors (NETs) of the pancreas. SPN often shows positive immunoexpression for cytokeratins, vimentin, CD10, CD56, neuron-specific enolase, progesterone receptor, cyclin D1, and CD99 in a perinuclear dot pattern. The SPN can also variably express synaptophysin. Hyaline blood cells show positive immunoexpression for α-1-antitrypsin. However, positive nuclear immunoexpression for β-catenin is also important for the diagnostic definition of SPN in the pancreas (Figure 4B).

### 3.5. Genetics Evaluation and Germline Testing

Five patients from our cases investigated the possibility of hereditary cancer predisposition. Their history and findings are described below:

Patient one—Female, history of bilateral asynchronous breast cancer at 39 years and 58 years, suprarenal adenoma at 43 years, lung adenocarcinoma at 47 years, and a SPN with a neuroendocrine component at 47 years (0.9 cm at first evaluation, 1.3 cm 3 months later). Molecular germline genetic testing revealed a pathogenic variant at *TP53* gene, c.1010G > A; p.(R337H), confirming the diagnosis of Li-Fraumeni syndrome.

Her family history was also extensive: one sister with breast cancer and sarcoma; four other sisters and one niece with breast cancer; one brother with prostate cancer; two nieces with CNS tumors; and two nephews with leukemia.

Patient two—Female, 41 y/o, SPN diagnosed during evaluation for hereditary cancer. She had a sister with breast cancer at 35 y, no molecular variants detected, a father with prostate cancer and meningioma, a paternal uncle with renal cancer, and another paternal uncle with colorectal cancer. The patient declined a test until after surgery.

Third patient—Female, with a sister and mother with pancreatic cancer. During a screening, MRI an asymptomatic SPN was detected at 48 y/o. Her additional family history included a maternal uncle with prostate cancer. No molecular tests were performed. Sister had a multigene panel without deleterious variants.

Fourth patient—Female, diagnosed with SPN, treated with surgery at 37 y/o, had multiple invasive breast cancer at 49 years, and had a maternal aunt with breast cancer. Multigene panel (91 genes, with copy number variant detection) showed a variant with uncertain significance (VUS) at *WRN* (NM_000553.4):c.2300C>G; p.(Thr767Arg), rs20182577.

Fifth patient—Male, diagnosed with SPN at 22 y/o, had a paternal aunt with pancreatic cancer at 66 y old, a paternal uncle with prostate cancer, parathyroid adenoma and basocellular carcinoma, no deleterious variants detected in a multigene panel, a paternal grandmother with pancreatic cancer at 86 years, and a maternal grandmother with rectal carcinoma.

## 4. Discussion

The SPN of the pancreas represents an unusual and rare tumor, with multiple literature reports around the world [5,12,13,14,15,16,17]. As observed by Papavramidis et al., approximately 90% of them were females, and 72% of the patients were included in the age group of 19–50 years old. Similar statistical data are seen in our study, whereas 90% of the patients are female and are included in the age range at diagnosis [6].

Clinical presentation varies, with 2/3 of our patients presenting dyspeptic or abdominal symptoms at diagnosis; meanwhile, the rest remained asymptomatic [17,18,19]. Around 15–40% of patients are diagnosed as asymptomatic [6,20].

On the subject of location, there are rare cases of extra pancreatic sites of disease such as mesocolon, retroperitoneum, liver, and omentum. The predominance of lesions is in the body and tail of the pancreas. Regarding tumors that are located in the pancreatic head, overall survival is shorter than those of other locations [20,21]. Typically, tumor markers and laboratory tests are within the normal range [21].

Regarding the cases illustrated here, the tumor in the diagnosis has a median range of around 3.2 cm. In one of the largest series ever published, with 718 evaluated patients, more than 80% of the patients presented tumors bigger than 5.0 cm [5]. This current sample presented in the report could represent a more contemporaneous observation of SPN, given that current diagnostic imaging is more accurate and sensitive, and given that the practice of routine imaging is more frequent nowadays [22,23,24].

Regarding the possible and most common findings in images: CT typically shows a single neoplasm with both solid and cystic components, with regions of hemorrhage and/or cystic degeneration and the presence of calcifications. Cystic components are more common in the central area, whereas solid areas are at the periphery of the mass, presenting contrast enhancement. Generally, the tumors are large with a mean transverse diameter of 8–11.5 cm. MRI improved our ability to diagnose SPN and better identify the classic well-circumscribed, encapsulated lesion with mixed signs of low/high intensity on T1 and T2 weighted images [25,26].

The role of surgery for SPN remains the foundation of the treatment, given that most tumors are diagnosed in a localized stage, have low malignant potential, and have a good prognosis [27]. The choice of surgical technique varies according to the location of the tumor in the pancreatic parenchyma [28,29].

The primary tumor resection confers a 95% chance of cure in 5 years [2,30]. There are reports of tumors with aggressive behavior and metastatic disease or relapse, but they are restricted to a minority of cases and do not correlate with tumor size [31,32]. None of our patients had obvious recurrence or distant metastasis. Considering the low malignant potential, Yepuri et al. reported in a recent systematic review a recurrence rate of 2.6%, only including studies with at least 5-y follow-up [29]. Male gender (OR 1.96), positive nodes (OR 11.9), positive margins (OR 11.1), and lymphovascular invasion (OR 5.5) were independent predictors of recurrence [29].

The surgical technique applied to 60% of our patients consisted of a distal pancreatectomy of the body-tail. Recently, few surgical groups have been studying less aggressive techniques, searching for an approach favoring organ preservation and contemplating procedures such as tumorectomy, enucleation, spleen preservation, central pancreatectomy, and even less aggressive lymphadenectomies. Nevertheless, there is no real consensus on the gold-standard method [4,33]. In our group, spleen preservation was comprehended in 53% of the patients, tumorectomy and enucleation was performed in four patients, and central pancreatectomy was performed in one patient, all with sustained benefit and no recurrence. Enucleation seems to be a valuable resource for cystic and neuroendocrine lesions, in general offering better intra-operative outcomes, shorter duration of procedures, and a lower risk of bleeding when compared to standard resections (30). Nevertheless, in some series, patients who underwent enucleation presented a higher risk of fistula as a complication (OR 1.46) [32].

In the context of molecular features, a common finding is the disrupted Wtn/β-catenin signaling pathways associated with cyclin D1 overexpression as well as the ErbB and GnRH signaling pathways. Also identified is the loss of heterozygosity for Harvey rat sarcoma viral oncogene (HRAS), mutation in exon 3 of CTNNB1, activated Hedgehog, androgen receptor, and epithelial mesenchymal transition (EMT)-coupled genes. These heterogeneities represent a peculiar neoplasm, thus also explaining the uncertain histogenesis. Prevalent positive expressions of vimentin, AAT, or NSE do not define a specific lineage, and the tumor cells do not regularly exhibit neuroendocrine/ductal/adenocarcinoma elements, giving rise to the hypothesis of a totipotent cell origin that would further differentiate [34]. The stroma may also present a hyalinized or myxoid appearance [35,36]. Immunohistochemistry is essential for diagnosis, as previously exemplified, because SPN often shows positive immunoexpression of CD99 in a perinuclear dot pattern [3,36,37].

Other molecular pathways have mixed evidence in the literature [38,39]. It is already shown that SPNs of the pancreas normally are *KRAS, TP53,* and *P16/CDKN2A/SMAD4* wild type. In this series of patients, we describe a case with confirmed Li-Fraumeni syndrome. To our knowledge, this is the first report evaluating germline genetic testing in SPN of the pancreas and describing this type of association. A variety of pancreatic neuroendocrine tumors are described in patients with Li-Fraumeni syndrome in some reports [40]. Furthermore, a different histology of pancreatic tumors consisting of adenocarcinomas are seven times more frequent in Li-Fraumeni patients [41,42].

Until now, there has been no strong evidence of hereditary syndromes associated with SPN, although almost no data about germline genetic testing in SPN is reported in literature. Of the five patients evaluated by a genetic counselor, two had a history of multiple tumors, with one of them further being diagnosed with Li-Fraumeni syndrome and two others with no relevant genetic findings having a family history of pancreatic adenocarcinoma.

Although surgery remains the mainstay of SPN treatment given the low metastatic potential, a greater understanding of this rare entity could be achieved with molecular testing. Cancer genetic risk assessment in SPN should be evaluated in more patients; however, nowadays no recommendation of germline genetic testing in SPN is formally indicated.

## Figures and Tables

**Figure 1 genes-13-01809-f001:**
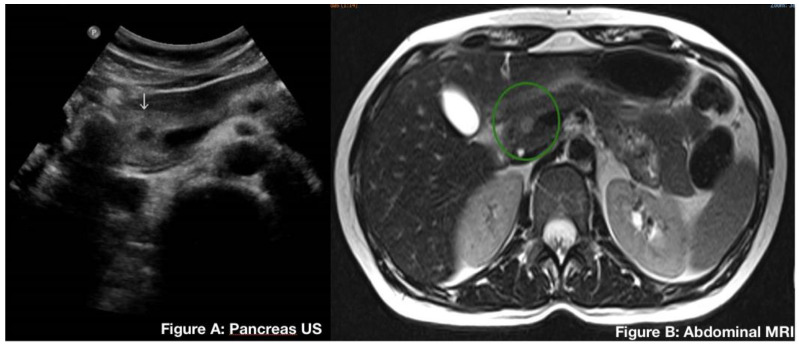
(**A**) (Arrow) Abdominal ultrasound showing a solid nodule of 0.9 cm on the head of the pancreas. (**B**) (Green circle) Abdominal magnetic resonance imaging (MRI) highlighting the same well circumscribed 0.9 cm nodule.

**Figure 2 genes-13-01809-f002:**
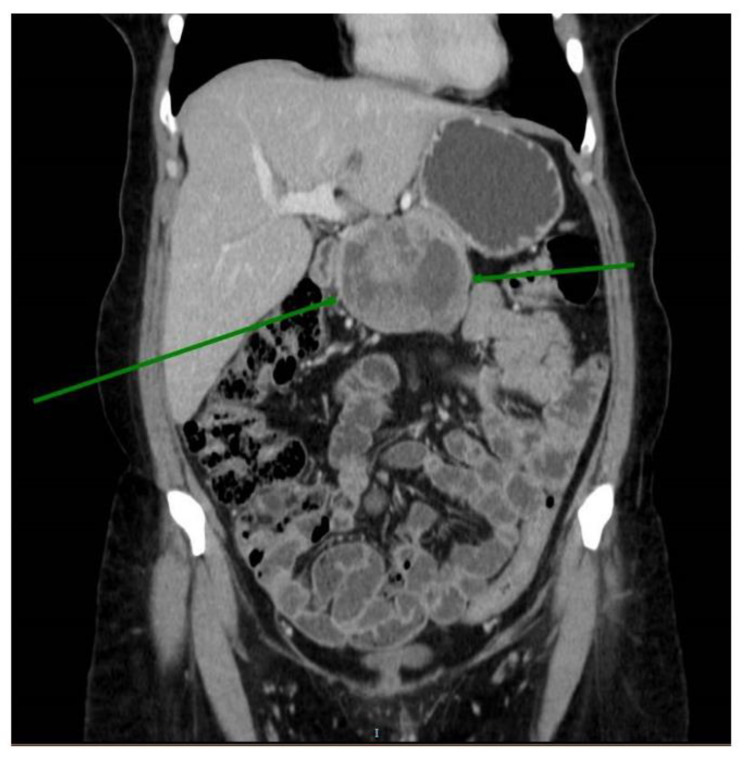
Abdominal computed tomography (CT) showing a greendemarcated solid cystic lesion of 7 cm in the head of the pancreas (Green lines).

**Figure 3 genes-13-01809-f003:**
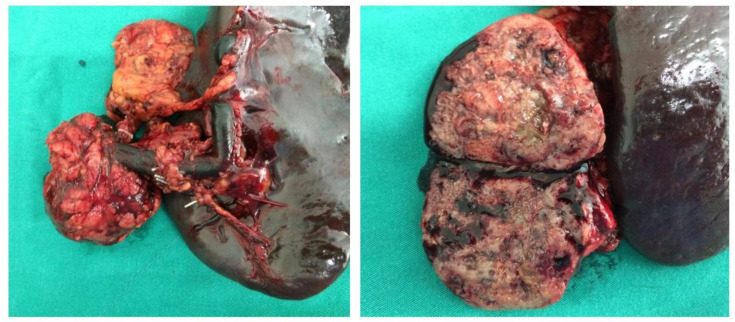
Macroscopic examination of the specimen of one of the reviewed cases, the product of a distal pancreatectomy: solid pseudopapillary neoplasm with areas of necrosis and hemorrhage, well delimited to the pancreatic tail.

**Figure 4 genes-13-01809-f004:**
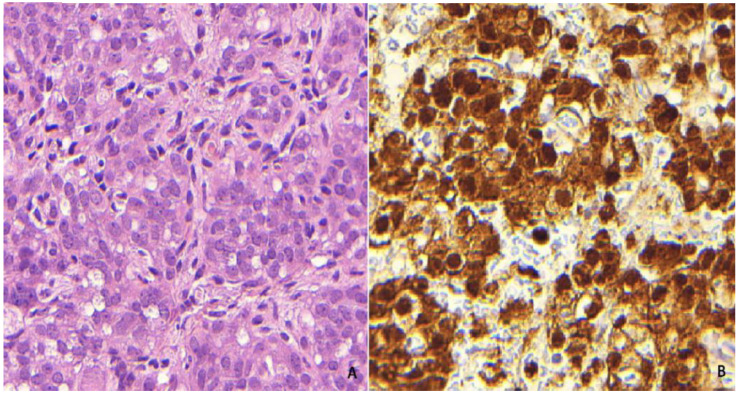
Solid pseudopapillary neoplasm immunohistochemical study. (**A**) Neoplasm composed of non-cohesive monomorphic cells in a solid pattern, with fibrovascular stroma (HE); (**B**) positive nuclear and cytoplasmic immunoexpression for β-catenin.

**Table 1 genes-13-01809-t001:** Demographic characteristics of patients.

Characteristics	Nº: 40 (100%)
Sex: - Female - Male	38 (95%) 2 (5%)
Age: Median (CI)	17 (10–49)
Symptoms: - Asymptomatic - Abdominal pain - Nausea and vomiting - Lose weight	16 (40%) 23 (57.5%) 3 (7.5%) 1 (2.5%)
Surgery: - Gastro-duodenectomy - Distal pancreatectomy - Nodulectomy - Central pancreatectomy	12 (30%) 23 (57.5%) 4 (10%) 1 (2.5%)
Tumor Size (cm): Median (CI)	3.6 (0.9–15)

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
