# Peer review of "Clinical Course, Genetic, and Immunohistochemical Characterization of Solid Pseudopapillary Tumor of the Pancreas (Frantz Tumors) in a Brazilian Cohort"

_genes, 2022, doi:10.3390/genes13101809_

Round 1

Reviewer 1 Report

Dear authors,

After the review process, I have several comments: as a research paper, you should include more numerical data in the abstract; the Materials and Methods section is too poor in details, with no references, and the statistical section should be present; the figures should have copyright or agreement to be publicly presented as open access.

Best regards!

Author Response

Manuscript ID - genes - 1851079

#Reviewer 1

First, we would like to thank you for your comments and relevant input regarding our manuscript. Please find below the reviewed version of the article, with your suggested changes incorporated to the best of our ability into the manuscript.

1: As a research paper, you should include more numerical data in the abstract; the Materials and Methods section is too poor in details, with no references, and the statistical section should be present; the figures should have copyright or agreement to be publicly presented as open access.

Response:

We tried to incorporate as much numerical data as possible in the abstract, and also the section materials and methods was rewritten and explained properly. The figures presented in the article are of open access with no necessity of copyright print.

Thank you for taking your time to revise our manuscript. We hope that these revisions make it acceptable for publication.

Yours sincerely,

Francinne Tostes, MD

Reviewer 2 Report

    Title: Current overview, genetics, and immunohistochemical characterization
of solid pseudopapillary tumor of the pancreas
(Frantz tumors)
Authors: Francinne T. Tostes, Parisina Fraga Dutra Cabral de Carvalho,
Raphael L.C. Araújo
, Rodrigo Chaves Ribeiro, Franz Robert Apodaca-Torrez,
Edson JoséLobo, Diogo Bugano Diniz Gomes, Donato Callegaro-Filho, Gustavo
Schvartsman
, Fernando Moura, Vladimir Schraibman, Alberto Goldenberg,
Fernanda Teresa de Lima
, Vanderlei Segatelli, Pedro Luiz Serrano Uson Junior

Comments: In this case series, the authors examine the clinical and pathologic data of
patients with solid pseudopapillary pancreatic neoplasms
(SPN) who have
undergone curative resection and the current mainstream
management of the
diagnosis and treatment of SPN.

This paper is not well structured, partially contradicts its own data, and does

not make clear whether it is a paper, a review, or a case report.

Entire text: The reader notices that different authors were involved here
because the style
varies and abbreviations are not used consistently (e.g.,
line 162). The text
should be restructured and homogenized.

Important major points are listed below:

1: Summary, p. 1, lines 26-29: This passage seems confusing:

(a) please do not switch to passive voice, but stick with "we", otherwise it

sounds like another paper is being discussed in the abstract, and

(b) there is confusion about whether this is a paper with completely
self-collected data, a
review paper, or case reports. This inconsistency runs throughout the paper and needs to be clarified. 2: The paper does not fit into the chosen journal because it does not primarily

deal with genetics.

3: Introduction, page 1, line 37: Please also cite Frantz's original 1959 reference
at this
point.

4: Introduction, page 2, line 48: Here and in line 72 it is emphasized that many

patients were symtomatic, whereas in the abstract it is emphasized that many

were asymptomatic. This change is consistent throughout the paper and it

should be made uniformly clear which fact is to be emphasized so as not to

confuse the reader.

5: Introduction, page 2, line 58: Please include a concluding paragraph about

what was done and why. This is also a good place to clarify whether the authors

collected the data themselves or just analyzed existing data.

M&M/Results/Discussion:

6: M&M, page 2: Was the research summarized here conducted by the authors
themselves or is
it purely a database analysis? Please clarify. If the studies were
self-conducted, they should be described in detail in M&M. 7: Results: If biomarkers are mentioned, their significance should also always
be briefly
explained, e.g., lines 75, 180-184, 245-253. 8: Results, page 3, lines 80-93: Own results should be described in the results.
The whole
paragraph consists of a list of sources that should be discussed in
the
discussion. Here, for example, it would be relevant why the results of the

references differ from the present work, e.g., for age and symptom proportion.

9: Results, page. 4, line 128: Please indicate the sex of the patient here, as in
line 121.
10: Results, p. 5, line 146: It needs to be checked if the references of the work
have shifted.
In the text Yepuri is mentioned as (18), according to the list (18) is
Tjaden.
11: Page 6, Figure 3: The images should be marked with characters and labeled.
The text
should explain exactly what is in the pictures. 12: English in general needs improvement (often sentences are linguistically

incorrect, but often words are simply missing or sentences are incomplete).

Author Response

Manuscript ID - genes - 1851079

#Reviewer 2

First, we would like to thank you for your comments and relevant input regarding our manuscript. Please find below the reviewed version of the article, with your suggested changes incorporated to the best of our ability into the manuscript.

This paper is not well structured, partially contradicts its own data, and does

not make clear whether it is a paper, a review, or a case report.

R: Thank you for your inquiry, we have now structured the manuscript as a research article.

Entire text: The reader notices that different authors were involved here
because the style varies and abbreviations are not used consistently (e.g.,
line 162). The text should be restructured and homogenized
.

R: Thank you for your inquiry, we have now homogenized the manuscript.

1: Summary, p. 1, lines 26-29: This passage seems confusing:

(a) please do not switch to passive voice, but stick with "we", otherwise it

sounds like another paper is being discussed in the abstract, and

R: Thank you for your inquiry, we have now corrected the text as suggested.

(b) there is confusion about whether this is a paper with completely
self-collected data, a review paper, or case reports. This inconsistency runs throughout the paper and needs to be clarified. 

R: Thank you for your inquiry, we have now structured the manuscript as a research article.

2: The paper does not fit into the chosen journal because it does not primarily

deal with genetics.

R: Thank you for your inquiry, this manuscript is submitted in the Journal section Human Genomics and Genetics Diseases, in the Special Issue Genotype-Phenotype Study in Disease. We describe in this study immunohistochemistry and genomics patterns in SPN. This article was previously accessed for suitability with editorial board.

3: Introduction, page 1, line 37: Please also cite Frantz's original 1959 reference
at this point.

R: Thank you for your suggestion, we have now included the reference.

4: Introduction, page 2, line 48: Here and in line 72 it is emphasized that many

patients were symtomatic, whereas in the abstract it is emphasized that many

were asymptomatic. This change is consistent throughout the paper and it

should be made uniformly clear which fact is to be emphasized so as not to

confuse the reader.

R: Thank you for pointing this out. We have now corrected the text.

5: Introduction, page 2, line 58: Please include a concluding paragraph about

what was done and why. This is also a good place to clarify whether the authors

collected the data themselves or just analyzed existing data.

R: Thank you for your suggestion, we have now included the suggested paragraph.

M&M/Results/Discussion:

6: M&M, page 2: Was the research summarized here conducted by the authors
themselves or is it purely a database analysis? Please clarify. If the studies were
self-conducted, they should be described in detail in M&M. 

R: Thank you for pointing this out. We have now corrected the text.

7: Results: If biomarkers are mentioned, their significance should also always
be briefly explained, e.g., lines 75, 180-184, 245-253. 

R: Thank you for pointing this out. We have now corrected the text.

8: Results, page 3, lines 80-93: Own results should be described in the results.
The whole paragraph consists of a list of sources that should be discussed in
the discussion. Here, for example, it would be relevant why the results of the

references differ from the present work, e.g., for age and symptom proportion.

R: Thank you for pointing this out. We have now corrected the text

9: Results, page. 4, line 128: Please indicate the sex of the patient here, as in
line 121. 

R: Thank you for pointing this out. We have now corrected the text

10: Results, p. 5, line 146: It needs to be checked if the references of the work
have shifted. In the text Yepuri is mentioned as (18), according to the list (18) is
Tjaden. 

R: Thank you for pointing this out. We have now corrected the references

11: Page 6, Figure 3: The images should be marked with characters and labeled.
The text should explain exactly what is in the pictures. 

R: Thank you for pointing this out. We have now corrected the legends. Both Figure 3 are the same tumor.

12: English in general needs improvement (often sentences are linguistically

incorrect, but often words are simply missing or sentences are incomplete).

R: Thank you for pointing this out. We have now corrected the text

Thank you for taking your time to revise our manuscript. We hope that these revisions make it acceptable for publication.

Yours sincerely,

Francinne Tostes, MD

Reviewer 3 Report

The authors uncovered genetics, and immunohistochemical characterization of solid pseudopapillary tumor of the pancreas, while providing an overview od the current knowledge.

Point to be addressed:

1. I would suggest to restructure the manuscript as follows:

P (Patient, population or problem)

Who or what is the patient, population or problem in question?

I (Intervention)

What is the intervention (action or treatment) being considered?

C (Comparison or control)

What other interventions should be considered?

O (Outcome or objective)

What is the desired or expected outcome or objective?

T (Time frame/treatment)

2. The authors employed a multivariable cox analysis: did they check for hazard's proportionality?

3. This reviewer personally misses some insights regarding biological background: as is now well known, tumors grow and evolve through a constant crosstalk with the surrounding microenvironment, and emerging evidence indicates that angiogenesis and immunosuppression frequently occur simultaneously in response to this crosstalk. Accordingly, strategies combining anti-angiogenic therapy and immunotherapy seem to have the potential to tip the balance of the tumor microenvironment and improve treatment response. Genetic alterations, especially the K-Ras mutation, carry the heaviest burden in the progression of pancreatic precursor lesions into pancreatic ductal adenocarcinoma (PDAC). The tumor microenvironment is one of the challenges that hinder the therapeutic approaches from functioning sufficiently and leads to the immune evasion of pancreatic malignant cells. Mastering the mechanisms of these two hallmarks of PDAC can help us in dealing with the obstacles in the way of treatment (please refer to PMID: 33918146 and expand the introduction/discussion sections). 

Author Response

Manuscript ID - genes - 1851079

#Reviewer 3

First, we would like to thank you for your comments and relevant input regarding our manuscript.

Point to be addressed:

  1. 1. I would suggest to restructure the manuscript as follows:

What is the intervention (action or treatment) being considered?

What other interventions should be considered?

What is the desired or expected outcome or objective?

R: Thank you for your inquiry, we have now restructured the manuscript as a research article.

  1. The authors employed a multivariable cox analysis: did they check for hazard's proportionality?

R: In this article we did not perform a multivariable cox analysis.

  1. This reviewer personally misses some insights regarding biological background: as is now well known, tumors grow and evolve through a constant crosstalk with the surrounding microenvironment, and emerging evidence indicates that angiogenesis and immunosuppression frequently occur simultaneously in response to this crosstalk. Accordingly, strategies combining anti-angiogenic therapy and immunotherapy seem to have the potential to tip the balance of the tumor microenvironment and improve treatment response. Genetic alterations, especially the K-Ras mutation, carry the heaviest burden in the progression of pancreatic precursor lesions into pancreatic ductal adenocarcinoma (PDAC). The tumor microenvironment is one of the challenges that hinder the therapeutic approaches from functioning sufficiently and leads to the immune evasion of pancreatic malignant cells. Mastering the mechanisms of these two hallmarks of PDAC can help us in dealing with the obstacles in the way of treatment (please refer to PMID: 33918146 and expand the introduction/discussion sections). 

R: We think that maybe some type of misunderstood is happening. Our manuscript is about Frantz tumor, not pancreatic adenocarcinoma. Biological mechanisms of Frantz tumor are described in the discussion.

We hope that these revisions make it acceptable for publication.

Yours sincerely,

Francinne Tostes, MD

Round 2

Reviewer 1 Report

No other comments.

Author Response

Thank you very much for the review.

Reviewer 2 Report

The authors have satisfactorily addressed the concerns raised in the original version. The revised version is significantly improved. No further concerns.

Author Response

Thank you very much for the review.